# Cascade Dual-Branch Deep Neural Networks for Retinal Layer and Fluid Segmentation of Optical Coherence Tomography Incorporating Relative Positional Map

**Da Ma**[*][1]                                                           DA_MA@SFU.CA
**Donghuan Lu**[*][2,1]                                                   DLA121@SFU.CA
**Morgan Heisler**[1]                                             MORGAN_HEISLER@SFU.CA
**Setareh Dabiri**[1]                                                   SDABIRI@SFU.CA
**Sieun Lee**[1]                                                          LEEAU@SFU.CA
**Gavin Weiguang Ding**[1]                                     GAVIN.W.DING@GMAIL.COM
**Marinko V. Sarunic**[1]                                               MSARUNIC@SFU.CA
**Mirza Faisal Beg**[1]                                              FAISAL_LAB@SFU.CA

[1] *School of Engineering Science, Simon Fraser University*
[2] *Tencent Jarvis Lab*

## Abstract

Optical coherence tomography (OCT) is a non-invasive imaging technology that can provide micrometer-resolution cross-sectional images of the inner structures of the eye. It is widely used for the diagnosis of ophthalmic diseases with retinal alteration such as layer deformation and fluid accumulation. In this paper, a novel framework was proposed to segment retinal layers with fluid presence. The main contribution of this study is two folds: 1) we developed a cascaded network framework to incorporate the prior structural knowledge; 2) we proposed a novel two-path deep neural network which includes both the U-Net architecture as well as the original implementation of the fully convolutional network, concatenated into a final multi-level dilated layer to achieve accurate simultaneous layer and fluid segmentation. Cross validation experiments proved that the proposed network has superior performance comparing with the state-of-the-art methods by up to 3%, and incorporating the relative positional map structural prior information could further improve the performance (up to 1%) regardless of the network.

**Keywords:** Retinal layer segmentation, Optical Coherence Tomography, Fully convolutional network, Relative positional map

## 1. Introduction

Optical coherence tomography (OCT) has been widely used to detect and monitor pathologies from retinal diseases. Anatomical and structural alteration measured from OCT images, such as layer thinning and fluid accumulation, are important signs for various types of retinal diseases (Hee et al., 1995; Joussen et al., 2010). However, manual segmentation of retinal layers and fluid is extremely time consuming, and suffers from inter-rater variability. Development of automatic segmentation tools can potentially help the physicians to achieve fast and accurate diagnosis.

---

[*] Contributed equally

The retinal layer segmentation methods can be categorized into two groups: 1) mathematical model based methods construct the models using prior assumptions of image structure, such as global shape regularization (Rathke et al., 2014) and graph (Lee et al., 2013) based methods. 2) pixel-wise classification based methods extracted pixel- or patch-wise features and feed to machine learning classifiers such as support vector machine (SVM) (Srinivasan et al., 2014) and deep learning based neural network (Roy et al., 2017). However, the performance of current available approaches are still behind the accuracy of human-rater's and new methods are needed for better segmentation accuracy.

In this study, a novel deep learning based framework is proposed to segment retinal layers with the presence of fluid. The major contributions of the study are: 1) Proposed a novel deep neural network for simultaneous retinal layer and fluid segmentation (LF-UNet), which combines the the U-Net (Ronneberger et al., 2015) and a original implementation of the FCN (Long et al., 2015) (referred to as FCN hereafter), and outperformed state-of-the-art methods in the cross validation experiments. 2) Proposed a novel framework with cascading networks to incorporate prior structural knowledge in a specific designed form, i.e, relative positional map. By calculating the relative positional map based on the segmentation of the first network and use it as additional channel of input for the second network, the performance of proposed approach was further improved regardless the network used in the framework.

## 2. Methods

Our framework for the segmentation of retinal layers and fluid consisted of two cascaded LF-UNet as displayed in Figure 1. First, the inner limiting membrane (ILM) and the Bruch's membrane (BM) were segmented by the first LF-UNet. Second, the relative positional map was calculated and used as an addition channel of input for the second LF-UNet to segment 6 retinal surfaces and fluid. A Random Forest classifier was trained in the last step to rule out false positive fluid regions as detailed in (Lu et al., 2019). The final outcome is the segmentation of both retinal fluid and 6 layer surfaces, including the ILM, the posterior boundary of nerve fiber layer (NFL), the posterior boundary of inner plexiform layer (IPL), the posterior boundary of outer plexiform layer (OPL), the IS/OS junction, and the Brunch's membrane.

### 2.1. Materials

The OCT images were acquired using a Zeiss Cirrus 5000 HD-OCT (Zeiss Meditec. Inc, Germany) which uses the OCT- micro-angiography complex algorithm (OMAG) with an A-scan rate of 68Khz. The 3x3mm pattern was used with a sampling rate of 245x245, which corresponds to a distance of $12.2\mu m$ between scanning locations. A total of 4 B-scans were acquired at each location. The A-scan depth of the system is 2mm with an axial resolution of $5\mu m$ and a transverse resolution of $15\mu m$. A total of 58 3D volumes were used in this study, 25 of which are acquired from diabetic patients who mainly exhibited intra-retinal fluid, and the remaining are acquired from healthy subjects. Each volume contains 245 B-scans, resulting a total of 14210 Bscans. We select and segmented retinal layers by referencing to their clinical relevance (Chiu et al., 2010). Five boundaries were first manually delineated: the inner limiting membrane (ILM); surface between the nerve fiber

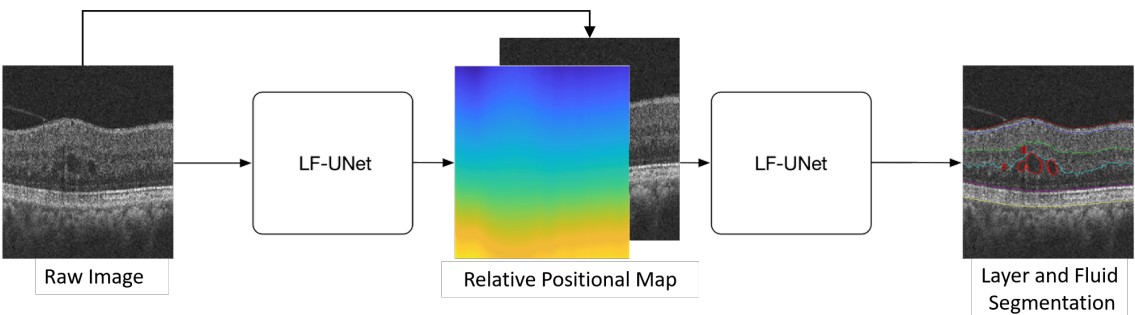

Figure 1: Flowchart of the proposed novel framework, comprising a cascade of two LF-UNet incorporating prior anatomical information, relative positional map within the retina, to achieve simultaneous layer and fluid segmentation.

layer(NFL) and the ganglion cell layer (GCL); surface between the inner plexiform layer (IPL) and the inner nuclear layer (INL); surface between the outer plexiform layer (OPL) and the outer nuclear layer (ONL); the inner/outer segment junction - surface between the Inner segment (IS) and outer segment (OS) of the photo receptor layer; the surface between the retinal pigment epithelial layer (RPE) and the bruch's membrain (BM). We then labelled the pixels between two retinal layer boundaries as the same class for training, instead of determining the layer boundaries, effectively converting the problem of boundary detection into tissue segmentation. 5 retinal layer structures were segmented, here referred to as ILM-NFL, GCL-IPL, INL-OPL, ONL-IS, OS-BM in the rest of this article. Retinal fluid can be categorized into different subtypes depending on their location in the retina [8]. In this study, given the limited number and size of the fluid regions, we regarded all the fluid as a single class.

## 2.2. Network Architecture

The network architecture of the proposed LF-UNet was a combination of the U-Net (Ronneberger et al., 2015) and the original implementation of FCN (Long et al., 2015). As illustrated in Figure 2, the U-Net and the FCN shared the same contracting path (the downward path on the left side). It consisted of 4 blocks which contained two convolution layers with kernel size $3 \times 3$ followed by an nonlinear activation function and a $2 \times 2$ max pooling layer with stride 2. The number of feature maps in each block were 64, 128, 256 and 512, respectively.

The expansive path was split into two parts: the U-Net part and the FCN part. In the U-Net part, the features extracted at each contracting block were concatenated with the features generated at the expansive block through skip-connections to provide high-resolution information. In the FCN part, the features of the contracting block and the expansive block with same resolution were added up as the input for the next block. The combination of these two networks harnesses their individual strengths, which leads to better segmentation. $2 \times 2$ up-convolution layer were used in both parts after convolutional layers.

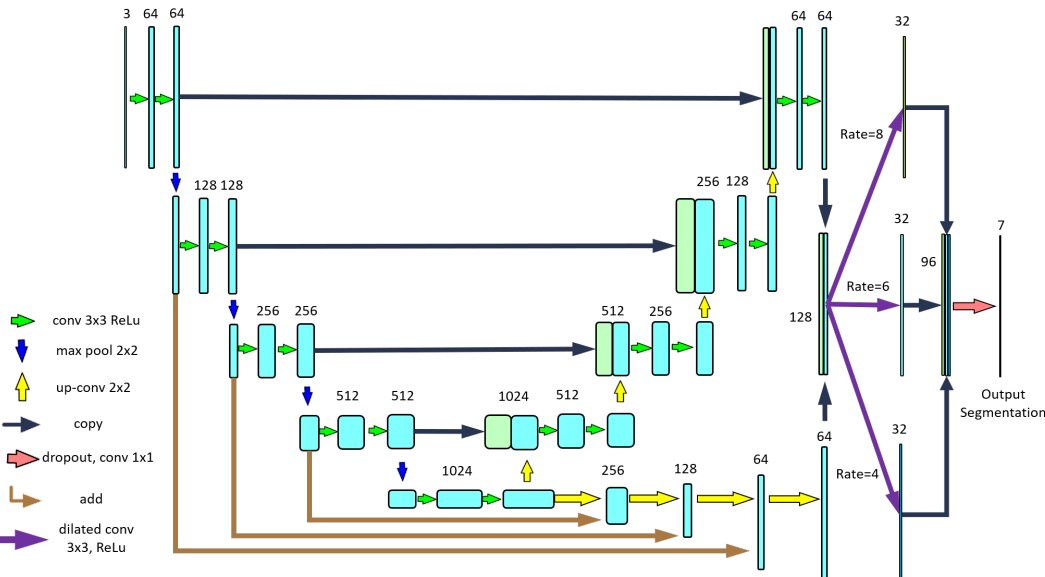

Figure 2: Network architecture of the LF-UNet, which is a combination of U-Net and Fully convolutional neural network. Each number above the cyan box represents the number of B-scans of the feature map.

The feature maps of the last convolutional layers in both parts were concatenated and fed to the three parallel dilated convolution layers (Yu and Koltun, 2015) followed by a single layer 1x1 convolutional network to predict the segmentation of different layers and the fluid. As some retinal layers occupied a large area, we used dilated convolutional layers instead of normal convolutional layers to increase the receptive field, enabling us to harness enough information from nearby layers. Using a large receptive field with normal convolutional layers would result in many more parameters thereby requiring more computational resources and potentially leading to overfitting. Conversely, dilated convolutional layers can enlarge the receptive field without increasing the number of parameters by skipping some units during convolution. All the activation functions used in the hidden layers were Rectified linear units (Relu), while the Softmax function was used for the output layer to map the output probability to $(0, 1)$.

### 2.3. Relative positional Map

The segmentation of the ILM and the BM was relatively easy due to their strong contrast compared with other layers and the background, while the segmentation of the remaining layers was more challenging due to their relative similar intensity patterns, especially for the posterior boundary of NFL, the posterior boundary of IPL and the posterior boundary of OPL, as shown in the right image of Figure 1. An important feature to determine the layer label of each pixel is its location in the retina. However, such features can hardly be captured by the network itself since the convolution kernel can only capture the information of nearby pixels. It has been shown that incorporating structural priors obtained from initial

segmentation of the entire retina can improve the network performance (Venhuizen et al., 2018). More recently, relative positional maps (Lu et al., 2019), a metrics more specific to the retinal anatomy, have proven to be a useful feature for the segmentation of multi-class fluid in the retina, and its potential usage in layer segmentation is worth exploring. For pixel $(x, y)$ in a B-scan, its intensity in the relative positional map is defined as:

$$I(x, y) = \frac{y - Y_1(x)}{Y_1(x) - Y_2(x)} \tag{1}$$

where $Y_1(x)$ and $Y_2(x)$ represent the $y$-coordinate of ILM and BM segmented by the first LF-UNet, respectively.

The relative positional map was concatenated to the B-scans as an additional channel of input for the second LF-UNet. Because the relative distance of background pixels above ILM was less than 0, while the relative distance of background pixels below ILM was larger than 1, they were labeled differently to avoid the confusion of network, resulting 8 mutually exclusive classes, background above ILM, ILM-NFL, NFL-IPL, IPL-OPL, OPL-IOS, IOS-BM, background below BM, and the fluid.

### 2.4. Loss function

The network was trained end-to-end with a loss function which consisted of two parts: the weighted Dice loss and the weighted logistic loss (Roy et al., 2017). The weighted Dice loss was defined as:

$$Loss_{Dice} = 1 - \frac{2 \sum_{x \in \Omega} \omega_l p_l(x) g_l(x)}{\sum_{x \in \Omega} p_l^2(x) + \sum_{x \in \Omega} g_l^2(x)} \tag{2}$$

where $\Omega$ represents the retinal region, $g_l(x)$ is the ground truth, $p_l(x)$ is the estimated probability of pixel $x$ belongs class $l$. $\omega_l$ is the weight associated with the number of pixels in different classes to resolve the imbalance among different layer regions and the fluid.

The weighted logistic loss is defined as below:

$$Loss_{log} = - \sum_{x \in \Omega} \omega(x) g_l(x) log(p_l(x)) \tag{3}$$

where $\omega$ is the weight associate with each pixel $x$.

In order to make the network more sensitive to boundary and retinal regions, the weight is designed as:

$$\omega(x) = 1 + \omega_1 I(|\bigtriangledown l(x)| > 0) + \omega_2 I(l(x) = L) \tag{4}$$

where $I$ represents an indicator function and $\bigtriangledown$ is the gradient operator. It is worth mentioning that $l$ is the label of pixel $x$ instead of its intensity, therefore a pixel with $|\bigtriangledown l(x)| > 0$ must be a boundary pixel based on its ground truth segmentation. $L$ represents the entire retina, including fluid and 5 layer regions. Following the suggestion from the original U-Net paper (Ronneberger et al., 2015), $\omega_1$ and $\omega_2$ were set as 10 and 5, respectively, and kept the same for all experiments.

The overall loss function was defined as:

$$Loss_{log} = \lambda_1 Loss_{Dice} + \lambda_2 Loss_{log} \tag{5}$$

where $\lambda_1$ and $\lambda_1$ were set as 0.5 and 1, respectively.

### 2.5. Optimization

Due to the limitation of GPU memory, the segmentation was performed on each 2D B-scan instead of the whole 3D volume. The adjacent B-scans (one before and one after the B-scan to be segmented) were also used for segmentation considering the consistency of the retinal layers and fluid, resulting a 500x245x3 matrix in the input of the first LF-UNet.

Two strategies were used during the training stage to make the proposed network less prone to overfitting. First, a dropout layer was inserted between the dilated convolutional layers and the $1 \times 1$ convolutional layer. The dropout ratio was set as 0.5 which means only half of the units were randomly retained to feed features to the last convolutional layer in the training stage. By avoiding training all units on every sample, this regularization technique not only reduced the chances of overfitting by preventing complex co-adaptations on the training data, but also reduced the amount of computation and improved training speed. Secondly, data augmentation was applied to create more training samples to improve the robustness and invariance properties of the network. Three types of image transformations were applied to augment the data - flip, rotation and scaling - with the maximum scaling ratio set as 0.5 and rotation degree set as $-25°$ to $25°$ in order to cope with cases where OCT images were acquired at peripheral side of the retina with large tilted angle.

Batch size was set as 3 due to the GPU memory limitation. The weight parameters for each layer were initialized with a uniform distribution while all bias started with 0 (Glorot and Bengio, 2010). Adaptive Moment Estimation (adam) optimizer was used for training with a fixed learning rate of $10^{-5}$ and the optimization was stopped if the training accuracy ceased to increase after 5 epochs.

## 3. Experiments and Results

The deep neural network was built with Tensorflow (Abadi et al., 2015), an open source deep learning toolbox provided by Google. All the experiments were run on NVIDIA P100-PCIE GPUs. To validate the ability of proposed framework, a 10-fold cross validation was performed on the 58 $3 \times 3mm$ volumes. To avoid the bias caused by using B-scans of the same volume in both training and testing, the volumes were partitioned on the patient level divided into the training set which contained the B-scans of 52-53 volumes, and the testing set which contained the B-scans from the rest of the volumes in each cross validation experiment. The segmentation performance was evaluated using both the Dice index and the surface distance for each B-scan. Performance for each layer and the fluid were measured separately, and the B-scans which did not contain fluid were discarded when measuring the performance of the fluid segmentation. Paired t-tests with multiple comparison controlled with False Discovery Rate (FDR) set to 0.05. The statistical comparison includes testing: 1) the effect of using 3-slice-channel compared to 1-slice-channel as input; 2) The effect of using cascaded network to incorporate spatial priors 3) The effect of network architecture change (U-net vs RelayNet vs LF-UNet). 4) Finally, the performance with and without fluid in the segmentation.

Table 1 and Figure 3 shows the comparison of the performance between the proposed framework and two state-of-the-art methods: the U-Net (Ronneberger et al., 2015) and the RelayNet (Roy et al., 2017). Two kinds of input, single B-scans and 3 consecutive B-scans, were tested with or without a relative positional map. To evaluate the effect of incorporating

Table 1: Dice index of different networks and inputs. 1U-Net-1Bscan represents only a single U-Net which used a single B-scan as input, while 2LF-UNet-3Bscan means 2 concatenated LF-UNets were used for segmentation with 3 adjacent B-scans as input, and the relative distance map calculated from the first LF-UNet was used as the additional channel of input for the second LF-UNet. Column 2 to 6 represents the segmentation accuracy for layers from top to bottom. Column 7 and 8 represents the Dice index for fluid before and after random forest classification. Noticing the 2LF-UNet-3Bscan has the best performance regarding the segmentation of fluid and most layers.

|  | ILM-NFL | NFL-IPL | IPL-OPL | OPL-IOS | IOS-BM | Fluid | RF-Fluid |
|---|---|---|---|---|---|---|---|
| 1U-Net-1Bscan | 0.8679 | 0.9315 | 0.9033 | 0.9153 | 0.9308 | 0.5011 | 0.3736 |
| 2U-Net-1Bscan | 0.8807 | 0.9454 | 0.9243 | 0.9411 | 0.9384 | 0.4871 | 0.4293 |
| 1U-Net-3Bscan | 0.8910 | 0.9451 | 0.9180 | 0.9271 | 0.9329 | 0.4761 | 0.3839 |
| 2U-Net-3Bscan | 0.9019 | 0.9526 | 0.9316 | 0.9458 | 0.9401 | 0.5066 | 0.4770 |
| 1RelayNet-1Bscan | 0.9032 | 0.9472 | 0.9208 | 0.9424 | 0.9432 | 0.4177 | 0.4079 |
| 2RelayNet-1Bscan | 0.9075 | 0.9500 | 0.9261 | 0.9472 | 0.9452 | 0.4676 | 0.4955 |
| 1RelayNet-3Bscan | 0.9236 | 0.9580 | 0.9355 | 0.9472 | 0.9451 | 0.4313 | 0.4299 |
| 2RelayNet-3Bscan | 0.9255 | 0.9593 | 0.9379 | 0.9517 | 0.9471 | 0.4471 | 0.3977 |
| 1LF-UNet-1Bscan | 0.9100 | 0.9531 | 0.9278 | 0.9466 | 0.9439 | 0.5014 | 0.4922 |
| 2LF-UNet-1Bscan | 0.9063 | 0.9507 | 0.9281 | 0.9484 | 0.9446 | **0.5132** | 0.5661 |
| 1LF-UNet-3Bscan | **0.9283** | 0.9610 | 0.9388 | 0.9509 | 0.9459 | 0.4674 | 0.4624 |
| 2LF-UNet-3Bscan | 0.9278 | **0.9612** | **0.9409** | **0.9526** | **0.9466** | 0.4985 | **0.5837** |

the fluid label towards the retinal layer segmentation accuracy, we also performed a set of additional experiments trained using the proposed network with only layer segmentation as ground truth labels. The evaluation results showed that, 1) Under the same condition, the proposed LF-UNet had better performance comparing with U-Net and RelayNet regarding the segmentation of both the retinal layers and fluid; 2) Using the two adjacent B-scans as addition channels of the input could significantly improve the segmentation accuracy regardless of the network architecture; 3) Under most circumstances, cascading networks to incorporate prior structural knowledge of retina could further improve the performance, suggesting the relative distance is a useful feature not only for fluid segmentation, but also for retinal layer segmentation.

Comparing with the layer segmentations, the fluid segmentation showed inferior performance with all three network architectures when using Dice, which is mainly due to the fact that patient with fluid of small size and dice index is sensitive to volume size by nature.

## 4. Conclusion and Discussion

In this paper, we have proposed a novel framework to automatically segment retinal layers as well as fluid in OCT images. A novel deep neural network, LF-UNet, was proposed and cross validation experiments proved that the proposed network outperformed state-of-the-art methods. Further experiments showed that by cascading two networks, incorporating structural prior knowledge using the relative positional map derived from the first network could improve the segmentation performance regardless of the network.

In our proposed network architecture, the two part of the expansion path utilize slightly different operations to propagate the features from the corresponding block in the contract-

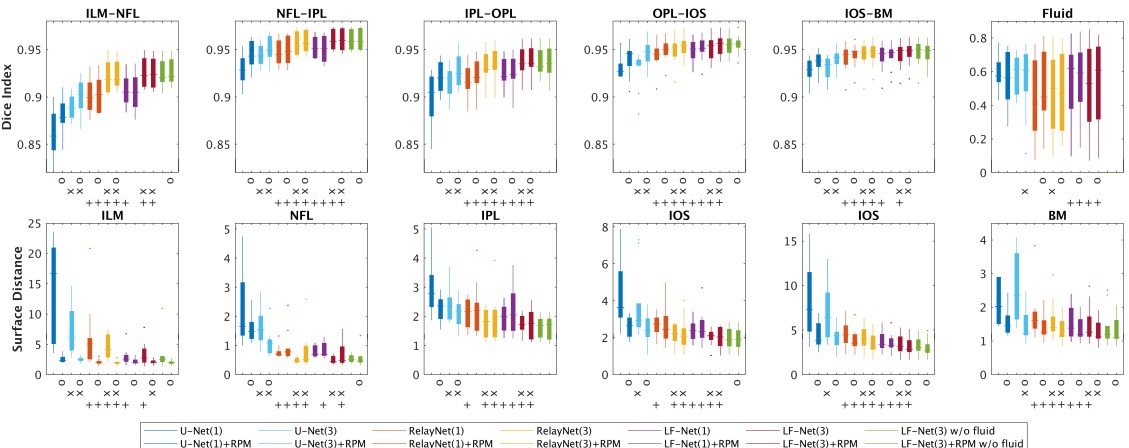

Figure 3: Performance Comparison among different experiments: - Top panel: Dice index; - bottom panel: surface distance Statistical label: - o: significant improvement of cascaded network training (with relative position map) over 1st round - x: significant improvement of 3-slice input channel over 1-slice input channel - +: significant improvement of network architecture

ing path: the U-Net path concatenate the feature maps through the "skip" connection, while the FCN path adding them together. This distinction is similar to the corresponding operations in RestNet(He et al., 2016) and DenseNet (Jegou et al., 2017; Huang et al., 2017), in which the concatenation and adding are introduced within a block of connecting layers. The key distinction is that, in this work, such "skipping connection" and "concatenation connection" are applied across the entire network covering features at all range, which can then been extracted by the last dilated convolution layers with different level of receptive fields. The significantly improved segmentation accuracy over the single-path approach in the U-Net demonstrated that the feature concatenation and summation retains complementary information embedded in the latent space.

As shown from our experimental comparison, the cascaded networks not only achieved significantly improved segmentation accuracy, especially on the boundary pixels, but also significantly reduced performance variations among test set, as demonstrated in the surface distance measure in Figure 3. This indicated the importance of incorporating anatomical prior information, even when automatically generated from the initial estimation, when performing segmentation tasks on medical image data. Similar conclusion has been demonstrated in different studies in which automatically generated spatial priors were incorporated in different form in a similar cascaded approach (Venhuizen et al., 2018).

Our experiment demonstrated that when training a neural network using 2D-convolution on 2D slices extracted from 3D volumes, incorporating adjacent slices as additional channels significantly improved the network performance. These additional channels potentially provide gradient information embedded in the local neighbourhood regions which are crucial

for segmentation tasks and can otherwise only be achieve through computational-expensive and training-data-hungry 3D-convolution-based volumetric segmentation.

In recognition of the above, we believe although this works is mainly focused on the retinal OCT layer and fluid segmentation, the results and conclusions drawn from the experiments could provides more generalizable insights contribute to the more broader domain of knowledge.

## Acknowledgments

This research received funding support from the Natural Sciences and Engineering Research Council of Canada (NSERC), Canadian Institutes for Health Research (CIHR), the Brain Canada Foundation, Alzheimer Society of Canada, the Pacific Alzheimer Research Foundation, Genome British Columbia, and the Michael Smith Foundation for Health Research (MSFHR).

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

## Appendix A. Supplementary Figure

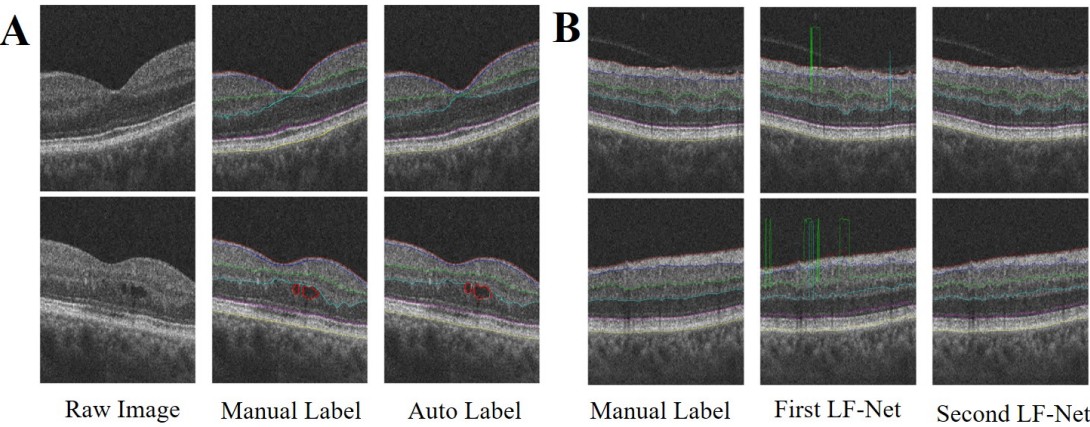

Raw Image     Manual Label     Auto Label       Manual Label     First LF-Net     Second LF-Net

Figure 4: Representative images of the segmentation results: A: Sample images demonstrating the simultaneous segmentation of retinal layer and fluid B: Sample images demonstrating the improved segmentation results when adding the relative positional map as additional input channel in the second cascaded LF-UNet (rightmost) compared to the results from the first LF-UNet (Middle)

