# OpenReview forum: "Cascade Dual-branch Deep Neural Networks for Retinal Layer and fluid Segmentation of Optical Coherence Tomography Incorporating Spatial Prior"
_MIDL.io/2020/Conference — MIDL 2020_

### Official Review · AnonReviewer1 · 2020-03-05
**Nice contribution but experiments missing**

**Rating:** 3
**Confidence:** 5
**Recommendation:** Poster

**Summary:**

* A two-stages deep learning approach for retinal layer and fluid segmentation in OCT.
* A combined architecture mixing a classical U-Net with dilated convolutions in the decoder.
* A relative distance map is used in the second stage to incorporate prior information regarding position of the layers.
* A Random Forest classifier is used after the network to remove false positive detections in the fluid area.
* Experiments on an in-house data set shows that the method outperforms two baselines (a U-Net and a RelayNet) in terms of average Dice.
* High performance (as measured by average Dice) for layer segmentation, still poor for fluid segmentation.
* Highly applicable to quantify layer properties in diseased patients suffering from macular diseases such as age-related macular degeneration, retinal vein occlusions or diabetic macular edema.

**Strengths:**

* The architecture proposed in the paper is novel in the sense that it integrates dilated convolutions in the decoder to capture multiresolution features.
* The incorporation of the relative distance map is not novel as it was already used in other medical imaging papers. However, it is novel in this specific application. The experiments in principle suggest that this prior knowledge aids the method to better segment the retinal layers in comparison with two other baselines.
* The two-stages approach outperforms two baselines in terms of average Dice.
* The problem is of interest for the retinal image analysis community.



**Weaknesses:**

* It is not clear if the improvements in performance can be attributed to the change in the U-Net architecture or to the incorporation of the two-stages approach. I would suggest the authors to include an additional experiment showing the performance of a single-stage LF-Unet model.
* The results are poorly presented on a table, in terms of average Dice, without including standard deviations and statistical tests showing the significance of the improvements in performance. The paper would be benefited by replacing Table 1 with a box plot showing the distribution of Dice values in the test set, and results of t-tests or Wilcoxon signed-rank tests comparing those distributions.
* No qualitative results are included in the paper. As a result, the reader won't be able to observe the impact of the low performance for fluid detection.

**Detailed Comments:**

The paper is using the format of LNCS instead of the Proceedings of Machine Learning. Take this into account!

Some other minor comments:

* "It is widely used for the diagnosis of ophthalmic diseases with retinal alteration, such as layer deformation and fluid accumulation". Remove the colon. Otherwise, you are referring to layer deformation and fluid accumulation as ophthalmic diseases instead of retinal alterations.
* "However, manual segmentation (...) suffers from inter-rater variability". Please, include a citation to support this statement.
* In second paragraph of Section 1: points 1 and 2 should be "Mathematical model based methods" (not *the*) and "Pixelwise classification based methods" (not *the*)
* Figure 1 should include a reference color map in the distance map.
* Indicate how many B-scans per volume are used.
* There are two different types of arrows in Fig. 2 using almost the same color (orange and brown). Please, change it so it's much easier to read.
* Shortcut connections should be referred as skip-connections, to be consistent with the literature and the original U-net paper.
* The phrase "The U-Net had good performance on the estimation of a coarse segmentation..." is a conclusion and should be moved away from the methodological section. I would suggest to include a few representative qualitative examples showing how the appearance of the structures is improved using the proposed model.
* Indicate for how many epochs the network was trained.
* Dice index is sometimes written with capital letter. Please, uniform this.

**Justification Of Rating:**

Although the contribution is definitely of interest for the MIDL audience, I'm afraid that the experimental part is missing some relevant experiments and is not well presented. I would suggest the authors to implement the changes I suggested to cover this point. If they do that, I would be more than happy to accept the paper.

**Paper Type:**

methodological development

**Questions To Address In The Rebuttal:**

* To highlight the contribution of the relative distance map, I would suggest the authors to include an additional experiment showing the performance of a single-stage LF-Unet model vs. the proposed approach (e.g. in the required box-plot).
* Replace Table 1 by a box-plot to see the distribution of Dice values (not just the mean).
* Discuss the statistical significance of the results.
* What happen if the Random Forest is not applied? Author mention in the paper that sometimes Dice index is reduced by this last approach because it can discard true positive pixels. Then: is it really necessary to apply it?
* It is still missing if the method improves thanks to the additional supervision of the position of the fluid. To proof this, I would suggest authors to perform an additional experiment training the proposed approach without the fluid class. If the layer results are better, then this extra supervision is aiding to improve performance. If not, then the major contribution of adding such a supervision is that you can get both the segmentation of the layers and the fluid simultaneously using a single model.
* From equation (1) we can see that the distance map can take negative values. Is it correct to refer to it as a distance map, then? Distances must be always positive.
* What is the motivation of using this combination of losses? How were fixed the the weights and other parameters of the loss? Were the baseline models trained using the same configuration?
* Aren't the -25º to 25º degrees rotations too strong for data augmentation? I'm afraid that those big angles might result in unrealistic retinal images.


**Special Issue:**

no

---

> ### Author Response · Authors · 2020-03-27
> **We would like to thank the reviewer for their encouraging and constructive comments. Our responses are listed below.**
>
>
> • Following the reviewer's suggestion, we have also substituted Table 1 with a box plot which now includes the results of all the statistical comparison: https://imgur.com/a/40bcZlU (with the written approval from MIDL organizer to share the image link).
>
>     • We have presented experimental results for both the single-stage and cascaded two-stage approach for all three different networks
>
>     • We have incorporated both the dice index (the higher the better) as well as the surface distance (the lower the better) as two separate metrics to measure the segmentation performance. Paired t-tests with multiple comparisons controlled with False Discovery Rate (FDR) set to 0.05.
>
> • We have also adjusted the result section about the performance of fluid segmentation with the additional "surface distance" metrics:
>  “Comparing with the layer segmentations, the fluid segmentations showed inferior performance with all three network architectures when using Dice, which is mainly because the small size of the fluid and dice index is sensitive to volume size by nature. However, when using the average surface distance as the metrics, the proposed method achieves less than 2 pixels error on average, significantly improved over the other two networks, indicating reliable fluid detection accuracy regardless of the fluid size.”
>
> • Our statistical analysis showed that the random forest for the fluid segmentation did not demonstrate statistically significant improvement. Since no additional added value has been presented with the random forest classifier, we have removed that in the experimental design to make the central message of the paper more concise and focused.
>
> • Following the reviewer's suggestion, we have performed additional experiments to exclude the fluid class in the training label, and the result is also shown in the new figure (the two green boxplots at the rightmost of each subplot of label dice scores and surface distance measurements).
>
> No significant difference was detected when comparing segmentation with or without the fluid label. Therefore, as an answer to the reviewer's question, the result did not show performance improvement with the presence of fluid label segmentation. That being said, we also acknowledge that the data with fluid presented are acquired from patients with relatively mild diabetic retinopathy, contains only small retinal fluid. Therefore, we're not excluding the potential improvement in the segmentation accuracy with the presence of fluid labels in other cases where a more severe level of retinal fluid presents. We will include these in the discussion in the final updated versions of the paper.
>
> • We thank the reviewer's suggestion about the nomenclature, and have changed the term "relative distance map" into "relative positional map (RPM)".
>
> • The combined loss function is based on the validated experiment from the cited study on the retinal layer and fluid segmentation (ReLayNet), which we have also included in the method comparison. (Roy, A. G. et al. (2017). ReLayNet: Retinal Layer and Fluid Segmentation of Macular Optical Coherence Tomography using Fully Convolutional Network. CoRR)
>
> • The values of the weights for the boundary and background pixels are based on the validated experiment from the original U-Net paper (Ronneberger, O. et al. U-Net: Convolutional Networks for Biomedical Image Segmentation, & Brox, T. (2015) MICCAI)
>
> • In terms of the parameters used for method comparison, we have ensured that the loss function is the same among all experiments for the U-Net, the RelayNet as well as the proposed LF-UNet.
>
> • We chose these large angles for data augmentation in order to cope with edge cases where OCT images were acquired at the peripheral side of the retina with large tilted angle, which were observed in the OCT datasets, which resulted in suboptimal segmentation results if the maximum rotation angle is set to too small. We will include this information in the final version of the paper.
>
> Detailed comments:
>
> • We have changed the paper format and have changed the correct format thanks for the reviewer's pointing out
>
> • The schematic diagram of the workflow has been updated with a colormapped relative positional map.
> • Same early stopping criteria are set for all experiments. However, due to early stopping, the number of epochs trained for each experiment are different. The average epoch across 10 cross-validations are:
> 14.5，13.75，14，13.35，12.7，12.1，13.55，12.8，13.9，15.6，17.05，17.45，19.3，19.45，18.45
>
> • There are 245 slices in each volume (sampling rate = 245x245). We've included these information in the updated version of the manuscript.
>
> • Finally, we would also like to thank the reviewer for their detailed comments on the correctness and potential improvement in the wording throughout the paper. We have made the corresponding changes following these comments.

---

### Official Review · AnonReviewer4 · 2020-03-09
**A cascaded deep network with relative distance map for fluid and retinal layer segmentation**

**Rating:** 3
**Confidence:** 4
**Recommendation:** Poster

**Summary:**

The paper proposes a cascaded deep network based on UNet architecture for the segmentation of retinal layers and fluids from OCT images.  First network segments the ILM and BM layer. Next, a relative distance map is computed from the output of the first network. The distance map with the input is fed to the final network to obtain a segmentation of 6 retinal surfaces and fluids. Finally, a Random Forest is trained for post-processing to remove the false fluid regions.

**Strengths:**

1. The main contribution of the paper is the cascaded network and the relative distance map. The
2.  The results justify the use of cascaded networks and in turn, evaluates the effect of the relative distance map.
3. The experiment to show the use of considering consecutive slices for segmentation is reported.
3. The authors compare their work with the two different CNN network to highlight the benefit of the relative distance map.

**Weaknesses:**

1. How expensive is distance map computation? and the RF post-processing step?
2. Is the training performed end to end or the cascade network is trained individually?
3. Why does your model tend to have more FP regions compared to RelayNet or UNet?
4. Is there thresholding applied for the output of deep network (UNet, RelayNet or LF-UNet) to get the fluid regions? If yes, could a different threshold select higher FP regions allowing Random Forest to improve the performance?

**Justification Of Rating:**

The paper is clearly written. The authors justify the use of the distance map with the cascaded network to segment retinal layers and fluids. The authors could have compared the results on publicly available datasets to better benchmark the paper against the cited reference.

**Paper Type:**

validation/application paper

**Special Issue:**

no

---

> ### Author Response · Authors · 2020-03-27
> **We would like to thank the reviewer for their comments and insightful thoughts about the potential improvements in our paper**
>
>
> The computation of the distance map from the segmented retinal layer is instantaneous. However, it requires the initial segmentation of the upper boundary (ILM) and the lower boundary (BM) of the retina, which is computed from the first run of the LF-UNet. Similarly, the random-forest (RF) post-processing step is fast, as long as the fluid segmentation is available, although the performance of the RF classification depends on the segmentation accuracy of the initial fluid segmentation (specifically the false-positive rate and the false-negative rate).
>
> At the moment, the two cascaded networks are trained separately, with the first network being trained to generate the initial layer segmentation for calculating the relative distance map, which is then used as an additional input channel for training the second network. However, we agree with the reviewer that an end-to-end approach, where the anatomical priors (relative distance map) could be updated within the training cycle would be very interesting to explore in the future.
>
> We speculate that one possible the reason that the FL-UNet showed improvement after Random Forest classifier while U-Net and ReLayNet don't might be due to the relatively low initial fluid detection rate, including both true-positives and false-positive of these two networks that prevent the random forest to effectively to improve the fluid segmentation accuracy.
>
> The segmentation output is from the pre-defined threshold (0.5) from the probability output of the softmax layer. We thank the reviewer for their insightful thoughts of inflating the threshold specifically for fluid labels thereby increase the false positive rate, thereby improving the effectiveness of the random forest. We would also like to share our thoughts on one potential limitation of this approach below：
>
> In those cases of false negatives, where there the region with ground truth fluid was not detected successfully originally, increasing the threshold would indeed potentially help to recover the false-negative areas.
> On the other hand, since the thresholding of the segmentation is performed on the pixel level (i.e. pixelwise classification), increasing the threshold would also have the effect of including additional pixels around the boundary area of the existing segmentation, effectively increasing the size of the segmentation labels.
>
> The main purpose of the random forest (RF) classification is to exclude those cases where there is no fluid exist in the ground truth data. In the cases where the originally predicted segmentation is false positive labels, the RF would not be able to remove the inflated segmentations labels, therefore causing the over-segmentation in the final results. We appreciate the reviewer for the constructive comments and will investigate this as additional hyperparameter optimization in the follow-up studies.

---

### Official Review · AnonReviewer2 · 2020-03-10
**The manuscript nicely presents a method to segment retinal layers and fluid from OCT scans.**

**Rating:** 4
**Confidence:** 5
**Recommendation:** Oral

**Summary:**

The manuscript presents two cascade networks for segmenting retinal layers and fluid from OCT scans, exploiting the concept of anatomical constraints introduced in [Lu et al., Medical Image Analysis 2019]. The addressed problem is relevant for the CAD community. The methodology presents some kind of innovation in how the anatomical constraints are computed.


**Strengths:**

The manuscript addresses a relevant and up to date problem, is well written and easy to follow. The proposed solution is novel and may be exploited also in close fields. Several experiments are performed to support the authors' investigation hypotheses.

**Weaknesses:**

Some methodological details are missing, hampering a proper understanding of the proposed methods. The experimental setup can be improved, for example by providing more performance metrics (right now, only the Dice similarity coefficient is provided). The survey of the state of the art can be improved by better highlighting limitations of current methods in the literature. The difference with respect to [10] and [7] should be highlighted better.

**Detailed Comments:**

The authors can find hereafter some specific comments to improve the overall manuscript readability.

Abstract
- The authors should be clearer when describing the second contribution (e.g., which are the benefits/ peculiarity of LF-UNet?)
- Consider including numerical results.

Introduction
- The first comment for the abstract applies also here. Instead of writing “Improved from the U-Net [9] and FCN [6], the proposed network outperformed state-of-the-art methods in the cross validation experiments.”, I would rather focus on the potential benefits of the LF-UNet architecture.
- I would not call the network in [6] as “FCN”, considering that it is a general word (for example, also U-Net is a FCN).

Methods
- The authors should include the random forest classifier in the workflow in Fig. 1.
- In Fig. 1, the authors should better specify how the raw image and relative distance map are combined.
- I suggest the authors, if space allows, to describe the clinical relevance of each of the segmented layers.
- The authors write “We regarded all the fluid as a single class”. This sentence is not clear to me. Do the authors mean that there are different kinds of fluid?
- In the caption of Fig. 2, the authors should avoid writing “which is a combination of U-Net and Fully convolutional neural network.”, considering that also U-Net is a fully convolutional neural network.
- Always in the caption of Fig. 2, what is the meaning of “number of B-scans of the feature map.”? Maybe the authors meant the number of feature maps?
- It is not clear, at this point of the manuscript, why the network starts with 3 channels. (It becomes clearer when reading Optimization). Please, fix.
- It would be nice to read something about the output dimension of the proposed network. For example, did the authors consider one output channel for each of the segmented structures? It seem so, with reference to Fig. 2. The authors should clarify and explain the rationale behind this choice.
- When describing the loss function, did the authors make any modification with respect to [10]?
- Did the authors perform augmentation on the fly (i.e., during training)?

Experiments and Results
- Why was the Dice similarity coefficient chosen as the only evaluation metric? Can the authors consider including also other metrics?
- How were the comparison methods chosen?
- The authors should be less generic when writing “Under most circumstances” / “For some experiments”.
- Dispersion measures could be added in Table 1.


**Justification Of Rating:**

I enjoyed reading the paper. The quality of the manuscript matches the MIDL requirements in terms of novelty and readability. The proposed methodology (i.e. the inclusion of anatomical priors in the segmentation pipeline) can be of inspiration for work in close fields.

**Paper Type:**

both

**Special Issue:**

yes

---

> ### Author Response · Authors · 2020-03-28
> **We appreciate the reviewer for their encouraging and constructive comments. Our detailed responses to each of the reviewer's comments are listed below:**
>
>
> ==================
> Abstract:
> ==================
> We have revised and improved the description of the second contribution of this study as below:
> 	" We proposed a novel two-path deep neural network which includes both the U-Net architecture as well as the original implementation of the FCN, concatenated into a final multi-level dilated layer to achieve simultaneous retinal layer and fluid segmentation (LF-UNet)."
>
> We have included numerical result as below
> 	"Cross-validation experiments proved that the proposed network has superior performance comparing with the state-of-the-art methods by up to 3%, and incorporating the relative positional map structural prior information could further improve the performance (up to 1%) regardless of the network."
>
> =====================
> Introduction
> =====================
> We have revised the corresponding section in the introduction where the "FCN"  was first mentioned as the "original implementation of FCN [6] (referred to as FCN hereafter)", to clarify any potential confusion, while respecting the definition in of original literature.
>
> We have revised the corresponding part of the Introduction:
> 	"The proposed LF-UNet combines three state-of-art network architecture in one single end-to-end trainable network:  U-Net [9], the original implementation fo FCN [6] and the dilated convolution [14]. Cross-validation evaluation demonstrated that the proposed netowrk and outperforms state-of-the-art methods".
>
> We have further elaborated the of the proposed network architecture in more detail, specifically on the difference between FCN and UNet as below:
> 	"In our proposed network architecture, the two parts of the expansion path utilize slightly different operations to propagate the features from the corresponding block in the contracting path: the U-Net path concatenate the feature maps through the "skip" connection, while the FCN path adding them together. This distinction is similar to the corresponding operations in RestNet and DenseNet, in which the concatenation and adding are introduced within a block of connecting layers. The key distinction is that, in this work, such "skipping connection" and "concatenation connection" are applied across the entire network covering features at all ranges, which can then been extracted by the last dilated convolution layers with different level of receptive fields. The significantly improved segmentation accuracy over the single-path approach in the U-Net demonstrated that the feature concatenation and summation retains complementary information embedded in the latent space."
>
> schematic diagram in Fig 1 will be updated thanks to the reviewer's suggestion
>
> We will include more clinical relevance of layer segmentation. We further refer to the following paper for the full clinical relevance for each segmented layers, in which our manual segmentations are also referred to.
> (Chiu, et al 2010. Automatic segmentation of seven retinal layers in SDOCT images congruent with expert manual segmentation. Optics Express)
>
> The legend of Fig 2 has been updated:  "number of B-scans or number of the feature maps"
>
> We have moved the introduction of 3-channel to earlier places to avoid potential confusion.
>
> Yes the output dimension of the network is indeed equal to the number of the segmented layers, which will be feed to the final softmax layer to produce the probabilistic prediction.
>
> The loss function used in this study is the same as the one introduced  in the reference [10]
>
> Yes, the data augmentation is performed in realtime during training with a randomized augmentation parameter.
>
> We have revised the commented sentence as below:  "Retinal fluid can be categorized into different subtypes depending on their location in the retina [8]. In this study, given the limited number and size of the fluid regions, we regarded all the fluid as a single class"
>   We have updated the Literature survey to highlight the current limitation in the literature.
>
> =======================
> Methods
> Experiments and results:
> =======================
>
> In response to reviewer's suggestion, we have introduced another performance metrics in addition to the "Dice similarity coefficient" - the surface distance, to focus on the evaluation on segmentation errors on boundary pixels, which is more relevant to the proposed medical application (retinal layer and fluid segmentation).
>
> As suggested by the reviewer, we have performed  Paired t-tests with multiple comparisons controlled with False Discovery Rate (FDR) set to 0.05, to compare:
> the effect of using 3-slice-channel compared to 1-slice-channel as input
> The effect of using the cascaded network to incorporate spatial priors
> The effect of network architecture change
> The effect of including fluid label in the segmentation
>
> We have produced a Figure which includes the statistical comparison results of both metrics
> https://imgur.com/a/40bcZlU (with the written approval from MIDL organizer to share the image link).

---

### Official Review · AnonReviewer3 · 2020-03-13
**A method for combined segmentation of fluid and retinal layers**

**Rating:** 2
**Confidence:** 5

**Summary:**

The authors propose a new deep learning framework for combined segmentation of retinal layers and fluid. They propose a new network architecture that merges a U-Net and a FCN, and apply it in a cascaded way to include prior knowledge about the location of the retina in the scan. The paper describes a combined repertoire of techniques (much of it appears to be based on prior work) that together leads to satisfactory performance. Performance on the task at hand seems to be the main goal of the authors, obtained through many ad-hoc and empirical decisions, which makes it hard to extract methodological contributions that have value in the broader field of knowledge.

**Strengths:**

- The paper is generally well written, well structured, and easy to follow.
- The paper contains a combination of many useful techniques (loss functions, network architecture, prior knowledge, post-processing) and the authors explain clearly how the combination of all of these can lead to increased performance.

**Weaknesses:**

- There is no clear scientific hypothesis or experimental validation behind much of the methodology. Many decisions seem to have been made ad-hoc (e.g. the combination of loss functions, value of the weights, the combination of the different layers in the proposed network architecture). This makes it much harder to identify the value of proposed novel methods outside the context in which they were developed.

- Going forward from the previous point: in such a setting with many manually tuned hyper-parameters, the comparison with other architectures/settings (as in Table 1) is a bit problematic. If hyper-parameters are optimized for the proposed network architecture/cascading setting, we cannot assume they are equally well-suited for other settings. This may bias the results towards the proposed setting, which is especially problematic since no statistical validation has been performed.

- The cascaded setting of incorporating prior structural knowledge is not completely novel. Although not identical, it is very similar to the approach in [1], which has not been mentioned in the references.


[1] Venhuizen, Freerk G., et al. "Deep learning approach for the detection and quantification of intraretinal cystoid fluid in multivendor optical coherence tomography." Biomedical optics express 9.4 (2018): 1545-1569.

**Detailed Comments:**

- Maybe introduce what LF in LF-UNet stands for?

- The data should be described in more detail. For example, how were the reference labels obtained? 25 are diabetic patients, what about the others? Please elaborate.

- The table with results is a bit misleading: 2RelayNet-3Bscan performs better on IOS-BM, while the bold numbers suggest 2LF-UNet-3Bscan is better.

'It is because random forest classification is better at rule out false positive regions, but unable to identify
the false negative regions - the fluid region missed by the network' => I believe this statement needs a reference or otherwise be supported by experimental evidence.


**Justification Of Rating:**

This paper mainly describes all the steps that the authors took to get to satisfactory performance. Although there are some interesting concepts that could be useful in other domains, it is very much tailored towards the specific application.

**Paper Type:**

methodological development

**Questions To Address In The Rebuttal:**

- The authors mention that optimization was stopped if training accuracy ceased to decrease. First, the definition of accuracy is not clear (pixel-accuracy averaged over all classes? Or Dice?). More importantly, this may not be optimal, as the network may be overfitting. Why not use an independent validation set to monitor performance?

- Please clarify the data split in the cross-validation, to ensure that data was partitioned on patient level, and the folds were completely independent.

- Please include some statistical validation of differences in performance between methods. For most layers, the differences between methods are very small.

- The discussion is quite limited. The only limitation that was mentioned is lack of performance on fluid segmentation, and the only suggestion is to collect more data. I believe the paper would benefit from including more insights that would convey how the proposed methodology contributes to the broader field of knowledge.

**Special Issue:**

no

---

> ### Author Response · Authors · 2020-03-27
> **We thank the reviewer for their constructive comments and have updated our paper accordingly. Our detailed response to the reviewer's comments are shown below**
>
> To answer the reviewer's question about the choice of hyper-parameters:
>
>     - The combined loss function is based on the validated experiment from the cited study [11] on the retinal layer and fluid segmentation (ReLayNet), which we have also included in the method comparison.
>     - The values of the weights for the boundary and background pixels are based on the validated experiment from the original U-Net paper [10]
>     -In our experiments, we have ensured that the loss function is the same among all experiments for the U-Net, the RelayNet as well as the proposed LF-UNet.
>
> Following the reviewer's suggestion, we have performed the statistical analysis. We have incorporated both the dice index (the higher the better) as well as the surface distance (the lower the better) as two separate metrics to measure the segmentation performance. Paired t-tests with multiple comparisons controlled with False Discovery Rate (FDR) set to 0.05.
>
> We have made a figure out of Table 1 which includes the results of all the statistical comparison: https://imgur.com/a/40bcZlU (with the written approval from MIDL organizer to share the image link).
>
> We have also included the recommended citation (Venhuizen2018), thanks for the reviewer's suggestion.
>
> The "training accuracy" is measured at batch level as the mean Dice index of the retinal layer and fluid labels across the whole batch. We agree with the reviewer's insights about the early stopping criteria for training the neural network. With regards to the current experimental setup, the result using monitoring the training performance showed a decent level of generalizability, and demonstrate segmentation performance difference between different experimental setup under the same hyperparameters. With that been said, we agree with the reviewer that, using independent validation sets would further prevent potential overfitting, and we are mentioning that in the discussion section of the updated version.
>
> We confirm that all the data are indeed partitioned on the patient level, where no slices from the same subject volume are split across the training and testing sets.
>
> Discussion: We thank the reviewer for their comments and suggestion on the discussion. We have added the following section below to include more insights that would convey how the proposed methodology contributes to the broader field of knowledge.
> ''
> In our proposed network architecture, the two parts of the expansion path utilize slightly different operations to propagate the features from the corresponding block in the contracting path: the U-Net path concatenate the feature maps through the "skip" connection, while the FCN path adding them together. This distinction is similar to the corresponding operations in RestNet and DenseNet, in which the concatenation and adding are introduced within a block of connecting layers. The key distinction is that, in this work, such "skipping connection" and "concatenation connection" are applied across the entire network covering features at all ranges, which can then been extracted by the last dilated convolution layers with different level of receptive fields. The significantly improved segmentation accuracy over the single-path approach in the U-Net demonstrated that the feature concatenation and summation retains complementary information embedded in the latent space.
>
> As shown from our experimental comparison, the cascaded networks not only achieved significantly improved segmentation accuracy especially on the boundary pixels, but also significantly reduced performance variations among test set. This indicated the importance of incorporating anatomical prior information, even when automatically generated from the initial estimation when performing segmentation tasks on medical image data. A similar conclusion has been demonstrated in different studies in which automatically generated spatial priors were incorporated in a different form in a similar cascaded approach (Venhuizen2018).
> ''
>
> We have included the following details in the paper following the reviewer's comments and suggestion:
> The LF-UNet stands for" U-Net-based Layer and Fluid segmentation network". We thank the reviewer for pointing this out, and has updated the text and explicitly define it.
>
> Subject details:
> A total of 58 3D volumes were used in this study, 25 of which are acquired from diabetic patients who mainly exhibited intraretinal fluid, and the remaining are acquired from healthy subjects.
> We have added the method to obtain the reference labels:
>
> Our statistical analysis showed that the improvement of the random forest for the fluid segmentation did not reach statistical significance. Since random forest classification is not the main focus of this paper, we have removed that in the experimental design.

---

> > ### Author Response · Authors · 2020-03-29
> > **Some additional explanation to further address reviewer's question about why "random forest classification is better at rule out false positive regions, but unable to identify the false negative regions - the fluid region missed by the network"**
> >
> > The random forest (RF) classifier is a postprocessing step applied to the segmented fluid labels which are detected by the proposed network. In another word, the "detected fluids" contain only true positives + false positives. Therefore, the RF is only capable of removing the false positives. For the false negatives - those fluid that are not detected by the proposed network (i.e. not segmented as fluid labels), they are not included in the input of RF classifier, therefore, cannot not be recovered.

---

### Comment · Area_Chair1 · 2020-04-01
**Interactive discussion**

I would like to thank the authors for their rebuttal. For the reviewers, could you please take some time to check the rebuttal and provide feedback? Thanks!

---

### Meta-Review · Area_Chair1 · 2020-04-06
**MetaReview of Paper13 by AreaChair1**

**Rating:** 3
**Recommendation For Accepted Papers:** Poster

**Metareview:**

There is a consensus among the reviewers (3 out of 4) in that this paper has merits to be accepted at MIDL. Reviewers generally consider that this works proposes a novel and valuable contribution to retinal image analysis community. While the experimental setting received some negative concerns during the initial reviewing process, the authors have positively addressed many of the comments. Taking into consideration both the initial and latest comments from the reviewers I recommend this paper for publication ('weak accept').

**Paper Type:**

both

**Special Issue:**

no

---

### Decision · Program_Chairs · 2020-04-11

Accept